# Microneedle manipulation of the mammalian spindle reveals specialized, short-lived reinforcement near chromosomes

**Pooja Suresh[1,2,3], Alexandra F Long[2,3,4], Sophie Dumont[1,2,3,4]\***

[1]Biophysics Graduate Program, University of California, San Francisco, San Francisco, United States; [2]Department of Cell and Tissue Biology, University of California, San Francisco, San Francisco, United States; [3]Department of Bioengineering and Therapeutic Sciences, University of California, San Francisco, San Francisco, United States; [4]Tetrad Graduate Program, University of California, San Francisco, San Francisco, United States

**Abstract** The spindle generates force to segregate chromosomes at cell division. In mammalian cells, kinetochore-fibers connect chromosomes to the spindle. The dynamic spindle anchors kinetochore-fibers in space and time to move chromosomes. Yet, how it does so remains poorly understood as we lack tools to directly challenge this anchorage. Here, we adapt microneedle manipulation to exert local forces on the spindle with spatiotemporal control. Pulling on kinetochore-fibers reveals the preservation of local architecture in the spindle-center over seconds. Sister, but not neighbor, kinetochore-fibers remain tightly coupled, restricting chromosome stretching. Further, pulled kinetochore-fibers pivot around poles but not chromosomes, retaining their orientation within 3 μm of chromosomes. This local reinforcement has a 20 s lifetime, and requires the microtubule crosslinker PRC1. Together, these observations indicate short-lived, specialized reinforcement in the spindle center. This could help protect chromosome attachments from transient forces while allowing spindle remodeling, and chromosome movements, over longer timescales.

**\*For correspondence:**
sophie.dumont@ucsf.edu

**Competing interests:** The authors declare that no competing interests exist.

## Introduction

The spindle is the macromolecular machine that segregates chromosomes at cell division. Mechanical force helps build the spindle, stabilize chromosome attachments (*Nicklas and Koch, 1969*), and ultimately move chromosomes (*Inoué and Salmon, 1995*). To perform its function, the mammalian spindle must generate and respond to force while maintaining a mechanically robust structure that can persist for about an hour. Yet, to remodel itself during mitosis, the spindle must also be dynamic, with its microtubules turning over on the order of seconds and minutes (*Gorbsky and Borisy, 1989*; *Saxton et al., 1984*; *Zhai et al., 1995*). How the spindle can be dynamic while also being mechanically robust remains an open question. While we know most of the molecules required for mammalian spindle function (*Hutchins et al., 2010*; *Neumann et al., 2010*), the spindle's emergent mechanical properties and underlying physical principles remain poorly understood. In large part, this is because of a lack of tools to probe the mammalian spindle's physical properties.

A key structural element of the mammalian spindle is the kinetochore-fiber (k-fiber), a bundle of 15–25 kinetochore-bound microtubules (kMTs) (*McEwen et al., 1998*) of which many reach the spindle pole (*McDonald et al., 1992*; *Rieder, 1981*). K-fibers generate force to move chromosomes (*Grishchuk et al., 2005*; *Koshland et al., 1988*) and provide connections to opposite spindle poles.

To do so, k-fibers must be robustly anchored and correctly oriented within the spindle. K-fibers make contacts along their length with a dense network of non-kinetochore microtubules (non-kMTs) (*Mastronarde et al., 1993*; *McDonald et al., 1992*), likely through both motor and non-motor microtubule binding proteins (*Elting et al., 2017*; *Kajtez et al., 2016*; *Vladimirou et al., 2013*). We know that the non-kMT network bridges sister k-fibers together (*Kajtez et al., 2016*; *Mastronarde et al., 1993*; *Tolić, 2018*; *Witt et al., 1981*), and that it can locally anchor k-fibers and bear load in the spindle's longitudinal (pole-pole) axis (*Elting et al., 2017*). Yet, how the dynamic spindle mechanically anchors k-fibers in space and time remains poorly mapped and understood. Specifically, we do not know if k-fibers are anchored uniformly along their length, to what structures they are anchored to, over what timescale this anchorage persists before remodeling is allowed, or more broadly how local forces propagate through the spindle's longitudinal and lateral axes. These questions are central to the spindle's ability to robustly maintain its structure, respond to force and ultimately move chromosomes.

We currently lack tools to apply forces with both spatial and temporal control to mammalian spindles. For example, laser ablation, commonly used to alter forces in the spindle, can locally perturb spindle structure, but lacks control over the duration and direction of ensuing force changes. Further, mammalian spindles cannot yet be reconstituted *in vitro*. To understand how the dynamic spindle robustly anchors k-fibers and ultimately map mammalian spindle mechanics to function, we need approaches to apply local and reproducible forces inside cells, with spatiotemporal control. Here, we adapt microneedle manipulation of the metaphase spindle in mammalian cells for the first time, combining it with live fluorescence imaging and molecular perturbations. We base our manipulation efforts on pioneering work in insect spermatocyte cells (*Nicklas and Koch, 1969*; *Nicklas et al., 1982*), newt cells (*Skibbens and Salmon, 1997*) and more recent work in *Xenopus* extract meiotic spindles (*Gatlin et al., 2010*; *Shimamoto et al., 2011*; *Takagi et al., 2019*).

Using this approach, we find that the mammalian mitotic spindle prioritizes the preservation of local structure in its center under seconds-long forces. We show that k-fibers can pivot around spindle poles but resist movement near chromosomes due to lateral and longitudinal reinforcement in the spindle center. We find that this reinforcement is specialized, only present near chromosomes, and short-lived with a lifetime of seconds. Finally, we show that this reinforcement is mediated by the microtubule crosslinker PRC1. Our work suggests a model for k-fiber anchorage that is local in both space and time: short-lived, local reinforcement isolates k-fibers from transient but not sustained forces in the spindle center. Thus, the spindle center can robustly maintain its connections to k-fibers and chromosomes, and yet remodel its structure and move chromosomes over minutes. Together, this study provides a framework for understanding how the spindle and other macromolecular machines can be dynamic yet mechanically robust to perform their cellular functions.

## Results

### Microneedle manipulation can exert local forces with spatiotemporal control on the mammalian spindle

To probe how the k-fiber is anchored in the mammalian spindle in space and time, we sought to mechanically challenge its connections to the rest of the spindle. Specifically, we looked for an approach to apply local forces on a k-fiber with the ability to control the position, direction and duration of force, in a system compatible with live fluorescence imaging to visualize spindle deformations. Based on work in insect meiotic spindles (*Lin et al., 2018*; *Nicklas and Koch, 1969*; *Nicklas et al., 1982*), we adapted microneedle manipulation to mammalian cells. We used PtK2 cells since they are molecularly tractable (*Udy et al., 2015*), flat, strongly adherent, and have a low number of chromosomes – helpful for pulling on individual k-fibers. We optimized several parameters to make this approach reproducible and compatible with cell health (detailed in Methods). We used a glass microneedle whose outer diameter was 1.1 ± 0.1 μm in the imaging plane, bent to contact the cell at a 90° angle, and fluorescently coated its tip to visualize its position. We connected the microneedle to a stepper-motor micromanipulator, and changed its x-y-z position either manually or with computer control. The latter ensured the smooth movements necessary to prevent cell membrane rupture, and to achieve reproducible microneedle motions from cell to cell.

To assess whether manipulation locally or globally perturbed dividing PtK2 cells, we imaged microtubules (GFP-tubulin), the microneedle (BSA-Alexa 555 or 647) and the membrane (CellMask Orange). We found that the microneedle deformed the membrane locally, rather than globally pressing the membrane against the whole spindle (*Figure 1A–B*). Consistent with local deformation, the overall cell height did not change upon manipulation (*Figure 1C*). The membrane appeared intact since the membrane contoured the microneedle during manipulation (*Figure 1B*) and the cell impermeable dye propidium iodide did not enter the cell during and after manipulation (*Figure 1—figure supplement 1*; *Nicklas et al., 1982*).

We used this approach to exert local, spatiotemporally controlled pulls on individual outer k-fibers in PtK2 GFP-tubulin metaphase cells (*Figure 1D*), and were able to deform their spindles. We pulled the k-fiber in the lateral direction (roughly perpendicular to the pole-pole axis) away from the spindle by 1.8 ± 0.4 µm for 11.9 ± 2.1 s (n = 7 cells) and 2.5 ± 0.2 µm for 60.5 ± 8.8 s (n = 23 cells) (*Figure 1E–F*). We imaged the spindle before, during and after the pull (*Figure 1G*, *Figure 1—video 1*) and found that the spindle returned to its original structure upon microneedle removal (*Figure 1H*). The spindle typically entered anaphase within 15 min of microneedle removal (*Figure 1G*, *Figure 1—video 1*), consistent with cell health. These observations indicate that we now have a local and reproducible approach to mechanically challenge the k-fiber's connections to the mammalian spindle over space and time.

## Pulling on kinetochore-fibers reveals the spindle's ability to retain local architecture near chromosomes under seconds-long forces

To probe how k-fibers are crosslinked to the spindle microtubule network, we examined the dataset where we manipulated the outer k-fiber over 11.9 ± 2.1 s (*Figure 1F*, red traces, n = 7 cells). This short timescale was chosen to probe the spindle's passive connections before significant remodeling had occurred: it is shorter than the lifetime of kinetochore-microtubules or detectable k-fiber growth or shrinkage (*Gorbsky and Borisy, 1989*; *Zhai et al., 1995*; *Figure 2—figure supplement 1*), and on the order of the half-life of non-kMTs (*Saxton et al., 1984*). We constructed strain maps (*Figure 2A*) to quantify the extent of deformation across the spindle in response to the manipulation. In principle, which structures move with the deformed k-fiber, and how much they move, could reveal the position and strength of anchorage within the spindle.

Upon manipulating the k-fiber over 12 s (*Figure 2B*, *Figure 2—video 1*, *Figure 2—figure supplement 2A*), we observed structural changes in the spindle that were local (*Figure 2C*, *Figure 2—figure supplement 2B*). The deformed k-fiber bent and deformations in the same spindle-half were only detectable within the first 5 µm from the microneedle (exponential decay constant = 0.44 µm$^{-1}$ (*Figure 2D*, *Figure 2—figure supplement 3A*)), suggesting weak mechanical coupling between neighboring k-fibers (*Elting et al., 2017*; *Vladimirou et al., 2013*). As a control, increasing crosslinking with a Kinesin-5 rigor drug (FCPT, n = 4 cells, *Groen et al., 2008*) led to a more gradual spatial decay of deformation (exponential decay constant = 0.25 µm$^{-1}$ (*Figure 2D*, *Figure 2—figure supplement 3B*)), with deformations propagating further through the spindle-half. This is consistent with the idea that crosslinking strength tunes anchorage within the spindle and thereby modulates its material properties (*Shimamoto et al., 2011*; *Takagi et al., 2019*). Together, these findings suggest that force propagation is dampened between neighboring k-fibers, which may effectively mechanically isolate them (*Matos et al., 2009*) and promote their independent functions.

Surprisingly, pulling on the k-fiber over this short timescale did not lead to an increase in interkinetochore distance (distance between sister k-fiber plus-ends, *Figure 2E,F*). Yet, we know that chromosomes relax and then stretch after k-fiber ablation near plus-ends over a similar timescale (*Elting et al., 2014*; *Sikirzhytski et al., 2014*), indicating that they are elastic on the timescale of these manipulations. Instead, the spindle shortened by 0.5 ± 0.1 µm in response to the manipulation (*Figure 2E,G*; *Gatlin et al., 2010*; *Itabashi et al., 2009*). This suggests that a structure other than the chromosome couples sister k-fibers across spindle halves on the seconds timescale. Consistent with this idea, the sister k-fiber, opposite the k-fiber being pulled, moved in towards the pole-pole axis by 5.8 ± 0.9° upon pulling (*Figure 2E,H*), preserving the angle between sister k-fibers (*Figure 2—figure supplement 4*). Together, this reveals that the spindle maintains local architecture around chromosomes against transient forces, instead adjusting its global architecture, and that sister k-fibers are tightly crosslinked to each other on the seconds timescale at metaphase.

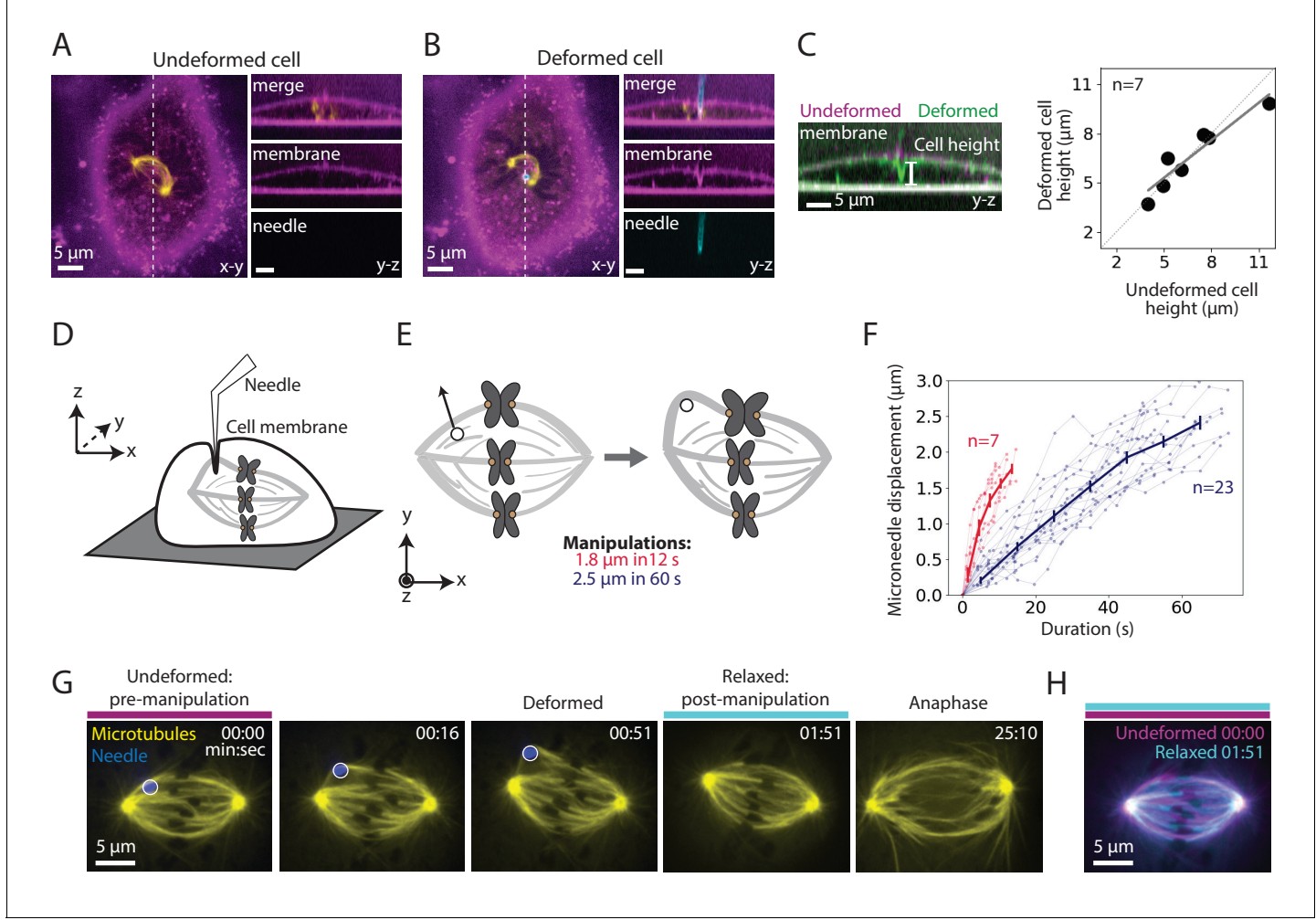

**Figure 1.** Microneedle manipulation can exert local forces with spatiotemporal control on the mammalian spindle. See also *Figure 1—figure supplement 1* and *Figure 1—video 1*. (A–B) Representative PtK2 cell (GFP-tubulin, yellow) and membrane label (CellMask Orange, magenta) (A) before (undeformed cell) and (B) during (deformed cell) microneedle (Alexa-647, blue) manipulation. x-y and y-z views displayed (left and right panels). y-z view taken along the white dashed line shown in the left panels. (C) Left: Overlay of the y-z view of the membrane labeled images before (undeformed, magenta) and during (deformed, green) microneedle manipulation, in order to compare membrane shape and cell height (white line) adjacent to the microneedle due to manipulation. Right: Cell height adjacent to the microneedle, measured using the membrane label, in its undeformed versus deformed state (n = 7 cells, Spearman R coefficient = 0.93, p=0.003, Pearson R coefficient = 0.94, p=0.002). Dashed line represents no change in cell height. Solid grey line is the linear regression fit to the data ($r^2$ = 0.88). (D) Schematic showing a very local deformation of the cell by the microneedle during manipulation, based on (A–C). (E) Schematic of the microneedle (black circle) manipulation assay used throughout this study, pulling (arrow) on a spindle's outer k-fiber for two different magnitudes and durations. (F) Microneedle displacement over time for two different manipulation datasets: 12 s (red, n = 7 cells) and 60 s (navy, n = 23 cells) pulls. Plot shows mean ± SEM. (G) Timelapse images of the representative response of a metaphase spindle in a PtK2 cell (GFP-tubulin, yellow), when its outer k-fiber is deformed by the microneedle (Alexa-647, blue, white circle) by 2.5 µm over 60 s. The spindle enters anaphase about 20 min after manipulation. Microneedle begins moving at 00:00 (first frame). Scale bar = 5 µm. Time in min:sec. (H) Overlay of the tubulin labeled images of the spindle (G) pre-manipulation (undeformed, magenta) and post-manipulation and microneedle removal (relaxed, cyan). The spindle's structure is similar pre- and post-manipulation, after correcting for spindle movement.

The online version of this article includes the following video, source data, and figure supplement(s) for figure 1:

**Source data 1.** This spreadsheet contains the height of a PtK2 cell when it is undeformed versus deformed by the microneedle (*Figure 1C*), and the microneedle displacement over time for both 12 s and 60 s manipulations (*Figure 1F*).

**Figure supplement 1.** Propidium iodide remains outside cells during microneedle manipulation.

**Figure 1—video 1.** Microneedle manipulation of a mammalian mitotic spindle at metaphase showing spindle relaxation and anaphase entry post-manipulation.

https://elifesciences.org/articles/53807#fig1video1

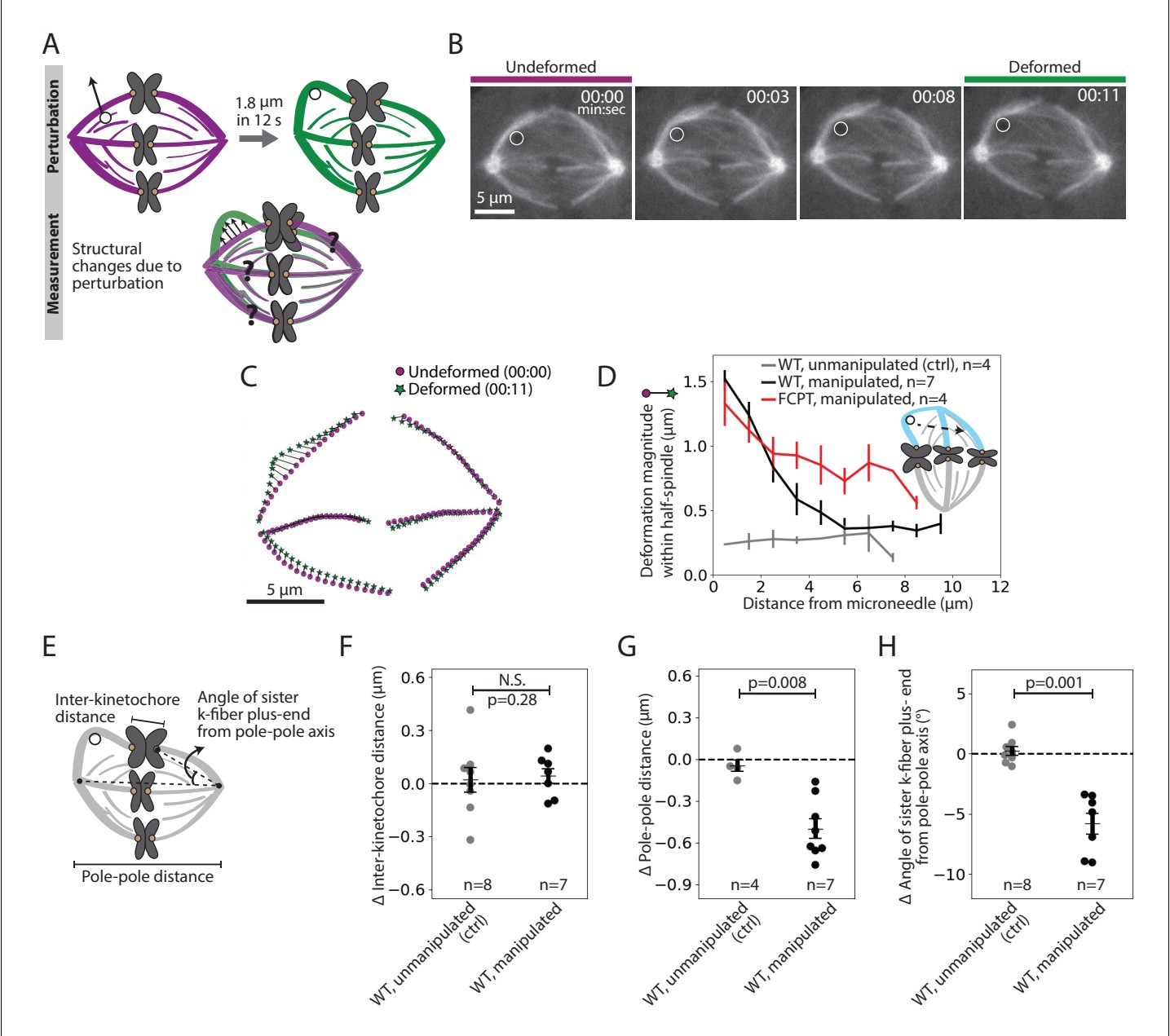

**Figure 2.** Pulling on kinetochore-fibers reveals the spindle's ability to retain local architecture near chromosomes under seconds-long forces. See also *Figure 2—figure supplements 1–4* and *Figure 2—video 1*. (A) Schematic of the assay to measure spindle deformation under local force: manipulation of the outer k-fiber for 12 s (perturbation) and generation of strain maps between undeformed (magenta) and deformed (green) spindles (measurement). (B) Timelapse images of a representative PtK2 metaphase spindle (GFP-tubulin, grey) during a 12 s manipulation, with microneedle position (white circle) displayed on images. Scale bar = 5 µm. Time in min:sec. (C) Strain map showing structural changes between undeformed (00:00, magenta circles) and deformed (00:11, green stars) spindles shown in (B), after correcting for spindle movement. Strain corresponds to the distance (black line) between magenta circles (undeformed spindle) and green stars (deformed spindle). (D) Magnitude of deformation in the structure (mean ± SEM) versus distance from the microneedle in unmanipulated WT (control, grey, n = 4 cells), manipulated WT (black, n = 7 cells) and manipulated FCPT-treated (positive control, red, n = 4 cells) spindles. (E) Schematic of the three measurements made in (F,G,H): Inter-kinetochore distance (measured between the manipulated k-fiber's and its sister's plus-ends), pole-pole distance, and angle between the sister k-fiber plus-end (opposite the manipulated k-fiber) and the pole-pole axis. (F) Change in inter-kinetochore distance in WT unmanipulated (control, n = 8 kinetochore pairs from 4 cells) and WT manipulated (between undeformed and deformed, n = 7 kinetochore pairs from 7 cells) spindles, measured over 12 s. There is no significant difference in the inter-kinetochore distance upon manipulation (p=0.28, Mann-Whitney U test). (G) Change in pole-pole distance in WT unmanipulated (control, n = 4 cells) and WT manipulated (between undeformed and deformed, n = 7 cells) spindles, measured over 12 s. Pole-pole distance decreases in manipulated spindles (p=0.008, Mann-Whitney U test). Plot shows mean ± SEM. (H) Change in angle of sister k-fiber plus-end with

*Figure 2 continued on next page*

*Figure 2 continued*

respect to the pole-pole axis, in WT unmanipulated (control, n = 8 k-fibers from 4 cells) and WT manipulated (between undeformed and deformed, n = 7 k-fibers from 7 cells) spindles, measured over 12 s. The sister k-fiber moves in towards the pole-pole axis in manipulated spindles (p=0.001, Mann-Whitney U test). Plot shows mean ± SEM.

The online version of this article includes the following video, source data, and figure supplement(s) for figure 2:

**Source data 1.** This spreadsheet contains the magnitude of deformation within the half-spindle versus the distance from the microneedle position in WT unmanipulated, WT manipulated and FCPT manipulated spindles manipulated over 12 s in PtK2 cells (*Figure 2D*).

**Figure supplement 1.** Kinetochore-fiber length does not change over 12 s manipulations.

**Figure supplement 1—source data 1.** This spreadsheet contains the change in k-fiber length in unmanipulated and manipulated spindles from 12 s manipulations in PtK2 cells.

**Figure supplement 2.** Additional example of a spindle manipulated for 12 s and its corresponding strain map.

**Figure supplement 3.** Estimating the exponential decay rate of spindle deformations over space.

**Figure supplement 3—source data 1.** This spreadsheet contains the magnitude of deformation within the half-spindle vs. the distance from the micro-needle position in WT and FCPT spindles manipulated over 12 s in PtK2 cells (same as *Figure 2D*).

**Figure supplement 4.** The angle between sister kinetochore-fibers is preserved in 12 s manipulations.

**Figure supplement 4—source data 1.** This spreadsheet contains the change in angle between sister k-fiber plus-ends in unmanipulated and manipulated spindles over 12 s, in PtK2 cells.

**Figure 2—video 1.** The spindle locally deforms under seconds-long forces.

https://elifesciences.org/articles/53807#fig2video1

## The deformed kinetochore-fiber's shape indicates specialized, short-lived crosslinking to the spindle near chromosomes

To probe the basis of the tight coupling between sister k-fibers, we measured curvature along the deformed k-fiber on the premise that it could inform on the spatial distribution of the effective underlying crosslinking (*Figure 3A*). In the k-fibers manipulated over 12 s, we observed high positive curvature at the position of the microneedle and, unexpectedly, a region of negative curvature near the chromosome (n = 4/7 cells) (*Figure 3—figure supplement 1A–B*). This configuration, more energetically costly than a single bent region, indicates that k-fibers are unable to pivot near their plus-ends, which is well suited to promote their biorientation.

To define the location and lifetime of these underlying connections, key to uncovering both mechanism and function, we repeated the above manipulation assay varying the position and duration of microneedle pulls. This time, we deformed the k-fiber by a larger magnitude of 2.5 ± 0.2 μm and for longer (60.5 ± 8.7 s, *Figure 1F*, navy traces, n = 23 cells) to accentuate the curvature profile (*Figure 3B*, *Figure 3—video 1*). Similar to the 12 s manipulations (*Figure 2F,H*), there was no increase in the inter-kinetochore distance and the sister k-fiber's plus-end moved in towards the pole-pole axis (*Figure 3—figure supplement 2A–C*). We observed negative curvature near the chromosome in 74% (n = 17/23) of the spindles, and near the pole in just 17% (n = 4/23) of the spindles (*Figure 3C–D*). This indicates that k-fibers can easily pivot around poles, as observed in insect cells (*Begg and Ellis, 1979*), but cannot do so around chromosomes. Often, manipulating the outer k-fiber exposed non-kMTs in contact with the k-fiber (*Figure 3B*, *Figure 3—figure supplement 1*); this revealed connections observed in electron microscopy (*McDonald et al., 1992*; *Nicklas et al., 1982*), that are harder to see with light microscopy. These non-kMT connections, observed close to the region of negative curvature (*Figure 3—figure supplement 3*), may contribute to reinforcing k-fibers in the spindle center.

To determine whether this chromosome-proximal reinforcement is mediated by uniform crosslinking all along the k-fiber length (Model 1) or specialized crosslinking near chromosomes (Model 2), we pulled the k-fiber at different distances from chromosomes (*Figure 3E*). Negative curvature was not correlated with microneedle position (*Figure 3F*), and was always observed between 1 and 3 μm from the chromosome regardless of where we pulled (*Figure 3G*). This suggests that the negative curvature we observe is not simply the result of local structural disruption caused by the microneedle's presence. Instead, it strongly supports a model whereby a specialized structure in the spindle center laterally reinforces k-fibers near chromosomes (Model 2).

To define the lifetime of this specialized reinforcement, we measured k-fiber curvature over time while we held the microneedle in place after manipulating for 60 s ('manipulate-and-hold', n = 5 cells). The negative curvature near chromosomes lasted for 18.8 ± 2.6 s before it was no longer

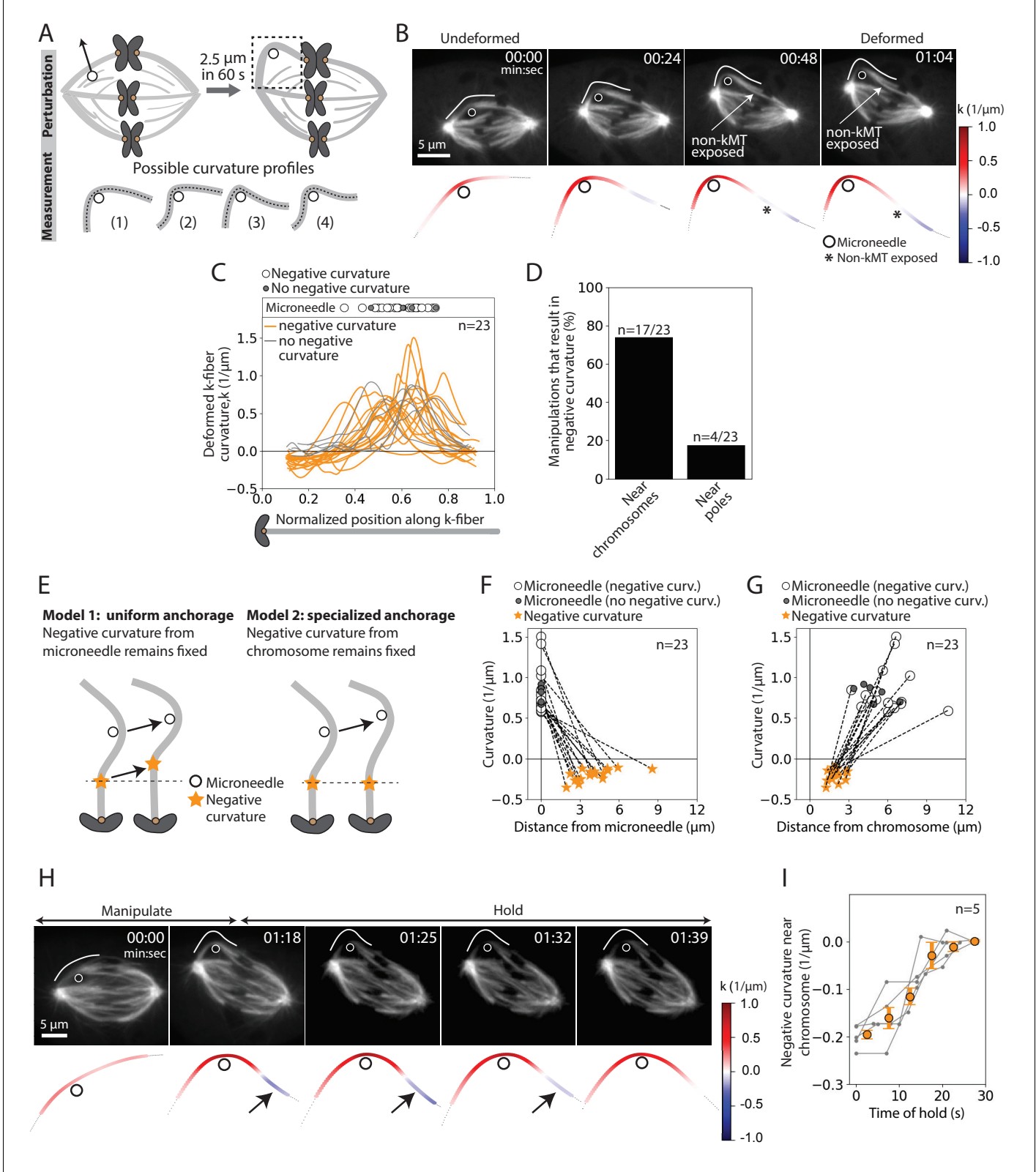

**Figure 3.** The deformed kinetochore-fiber's shape indicates specialized, short-lived crosslinking to the spindle near chromosomes. See also *Figure 3— figure supplements 1–3* and *Figure 3—videos 1* and *2*. (**A**) Schematic of the assay to probe the physical basis of k-fiber anchorage in the spindle: manipulation of the outer k-fiber for 60 s and quantification of local curvature along its length. The absence of k-fiber negative curvature (1) would suggest pivoting at poles and chromosomes. K-fiber negative curvature at poles (2) or chromosomes (3) or at both (4) would suggest it is laterally

*Figure 3 continued on next page*

Figure 3 continued

anchored there and prevented from pivoting. (B) Top: Timelapse images of a representative PtK2 metaphase spindle (GFP-tubulin, grey) during a 60 s manipulation, with microneedle position (white circle) and traced manipulated k-fiber (white) displayed on the images. Scale bar = 5 µm. Time in min: sec. Bottom: Curvature mapped along highlighted k-fiber for each time point in the top panel (blue, negative curvature; red, positive curvature). This manipulation can expose contact points (asterisk) between the k-fiber and non-kMTs. (C) Local curvature of deformed k-fibers for normalized positions along the k-fiber (n = 23 cells). Most k-fibers exhibit negative curvature near the chromosome (orange), and a few show no negative curvature (grey) near the chromosome. Few k-fibers also show negative curvature near poles. Scatter plot of microneedle positions shown above (inset). (D) Percentage of k-fiber curvature profiles with negative curvature less than −0.1 1/µm, proximal to chromosomes (n = 17/23 cells) and the pole (n = 3/23 cells). (E) Schematic of two possible outcomes of manipulating the outer k-fiber at different locations along its length: either the negative curvature position (orange star) remains fixed relative to the microneedle (black circle) position (uniform anchorage along the k-fiber, Model 1) or remains fixed relative to the chromosome (specialized, non-uniform anchorage near chromosome, Model 2). (F–G) Position of the curvature maxima (microneedle, white circle) and curvature minima (negative curvature, orange star) (F) measured from the microneedle position (n = 23 cells), and (G) measured from the chromosome (n = 23 cells). Dashed lines connect the maxima (microneedle) and minima (negative curvature) for a given manipulation. The negative curvature position is tightly distributed near chromosomes, regardless of the microneedle's position, supporting a specialized crosslinking model (Model 2, E). Plot also shows microneedle positions of manipulations that do not result in negative curvature (grey circles). (H) Top: Timelapse images of a PtK2 metaphase spindle (GFP-tubulin, grey) manipulate-and-hold experiment to probe the timescale of k-fiber reinforcement in the spindle center, performing a 60 s manipulation and then holding the microneedle (white circle) in place to measure when the negative curvature in the manipulated k-fiber (white trace) disappears (1:39, 21 s after 1:18 hold started). Scale bar = 5 µm. Time in min:sec. Bottom: Curvature mapped along highlighted k-fiber for each point in the top panel (blue, negative curvature; red, positive curvature). Negative curvature (black arrow) disappears over 21 s of holding time. (I) Curvature minima near chromosome as a function of time the microneedle has been held in place (n = 5 cells). Negative curvature disappears after holding for 20 s. Plot shows mean ± SEM (orange).

The online version of this article includes the following video, source data, and figure supplement(s) for figure 3:

**Source data 1.** This spreadsheet contains the local curvature along k-fibers manipulated over 60 s in PtK2 cells (*Figure 3C*), the positions of the microneedle and negative curvature with respect to the plus-end and the microneedle as well as their respective curvature values (*Figure 3F–G*), and the negative curvature near chromosomes during the hold time of the 'manipulate-and-hold' assays (*Figure 3I*).

**Figure supplement 1.** Deformed kinetochore-fibers exhibit negative curvature in 12 s manipulations.

**Figure supplement 1—source data 1.** This spreadsheet contains the local curvature along k-fibers manipulated over 12 s in PtK2 cells (*Figure 3—figure supplement 1B*).

**Figure supplement 2.** Tight coupling between sister kinetochore-fibers in 60 s manipulations.

**Figure supplement 2—source data 1.** This spreadsheet contains the change in inter-kinetochore distance (*Figure 3—figure supplement 2B*) and angle of sister k-fiber plus-end from the pole-pole axis (*Figure 3—figure supplement 2C*) in unmanipulated and manipulated spindles over 60 s.

**Figure supplement 3.** Non-kinetochore microtubule contacts distributed close to observed negative curvature.

**Figure supplement 3—source data 1.** This spreadsheet contains the position of negative curvature from the k-fiber plus-end, position of non-kinetochore microtubule contact from the k-fiber plus-end, and the distance between them.

**Figure 3—video 1.** Microneedle manipulation of a kinetochore-fiber reveals pivoting around poles and local reinforcement near chromosomes.
https://elifesciences.org/articles/53807#fig3video1

**Figure 3—video 2.** Manipulate-and-hold assay reveals that local reinforcement near chromosome has a 20 s lifetime.
https://elifesciences.org/articles/53807#fig3video2

---

detectable, likely reflecting the lifetime of the underlying connections (*Figure 3H–I*, *Figure 3—video 2*). Together, these findings indicate the presence of short-lived, non-uniform reinforcement of the k-fiber near chromosomes that is stable enough to preserve spindle structure over short timescales, but sufficiently dynamic to allow spindle remodeling over long timescales.

## The microtubule crosslinker PRC1 mediates the specialized and short-lived kinetochore-fiber reinforcement near chromosomes

We next sought to determine the underlying molecular basis for this specialized, short-lived reinforcement near chromosomes. We hypothesized that the microtubule crosslinker PRC1 plays this role based on its localization in the spindle center during metaphase (*Mollinari et al., 2002*), where antiparallel microtubules are present, and its proposed role in linking sister k-fibers at metaphase and anaphase (*Jiang et al., 1998*; *Kajtez et al., 2016*; *Mollinari et al., 2002*; *Polak et al., 2017*). Using immunofluorescence, we first asked if PRC1 localization in PtK2 cells correlated with the location of this specialized reinforcement region (*Figure 4A*). We found that PRC1 enrichment spanned a region of 6.9 ± 0.3 µm (n = 6 cells) in the spindle center (*Figure 4B*). This maps well to the expected location of a specialized crosslinker, spanning the inter-kinetochore region (~2 µm), and the region of mechanical reinforcement near chromosomes (1–3 µm along each sister k-fiber).

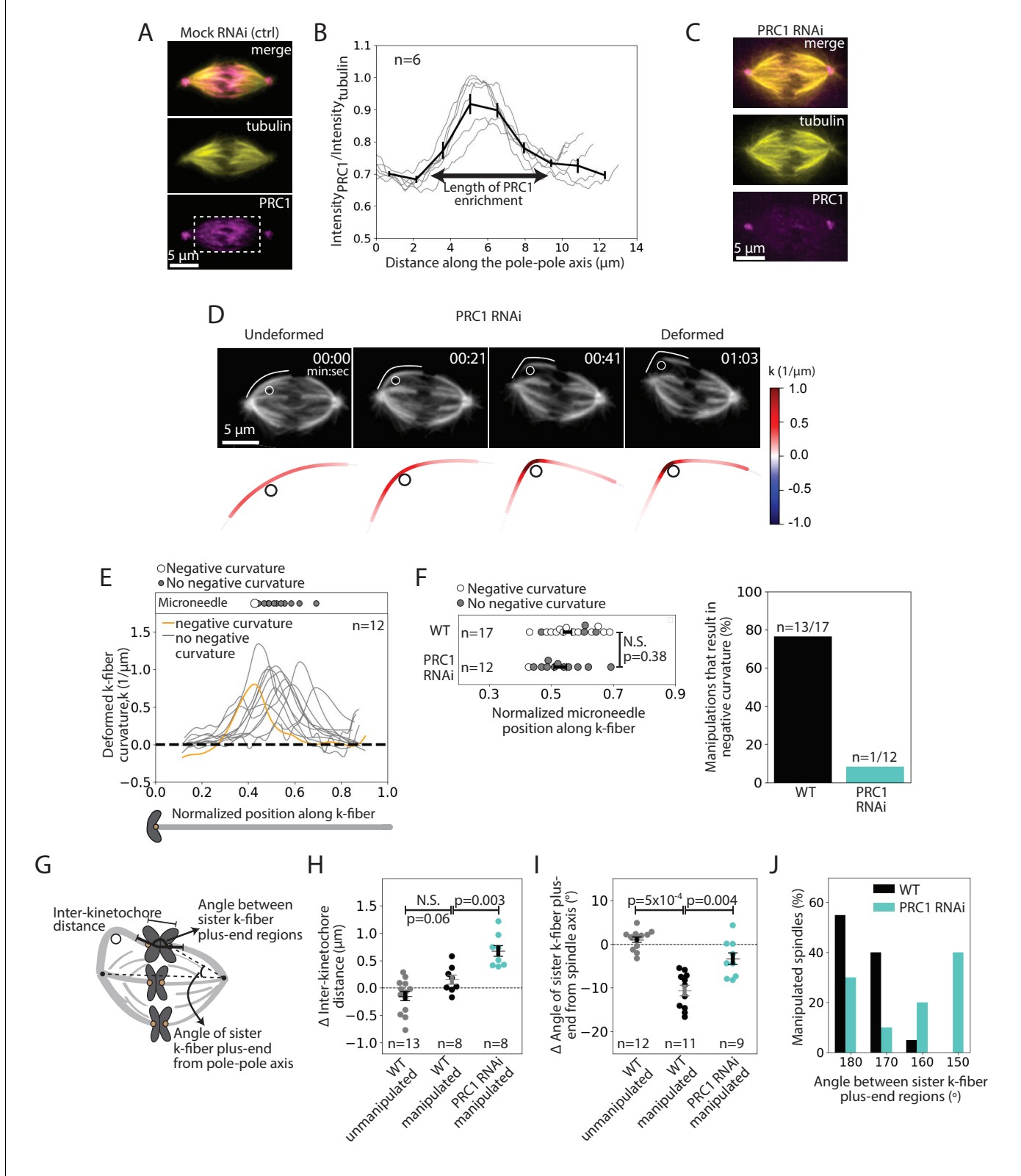

**Figure 4.** The microtubule crosslinker PRC1 mediates the specialized and short-lived kinetochore-fiber reinforcement near chromosomes. See also *Figure 4—figure supplements 1–2* and *Figure 4—video 1*. (A) Immunofluorescence images of a representative PtK2 mock RNAi (control) metaphase spindle showing where PRC1 is localized in the spindle (tubulin, yellow; PRC1, magenta). White box (bottom panel) shows the region in which intensity (B) was quantified. Scale bar = 5 μm. (B) Fluorescence intensity ratio of PRC1 to tubulin along the pole-pole axis (n = 6 cells), showing PRC1 localization

*Figure 4 continued on next page*

*Figure 4 continued*

in the spindle center. Plot shows mean ± SEM. (**C**) Immunofluorescence images of a representative PtK2 PRC1 RNAi metaphase spindle (tubulin, yellow; PRC1, magenta), showing PRC1 depletion. Scale bar = 5 μm. (**D**) Top: Timelapse images of a representative PtK2 metaphase PRC1 RNAi spindle (GFP-tubulin, grey) during a 60 s manipulation, showing microneedle position (white circle) and traced manipulated k-fiber (white). Scale bar = 5 μm. Time in min:sec. Bottom: Curvature mapped along traced k-fiber for each point in the top panel (blue, negative curvature; red, positive curvature), showing the absence of negative curvature near chromosomes without PRC1. (**E**) Local curvature of deformed k-fibers for normalized positions along the k-fiber (n = 12 k-fibers in 12 cells). Most k-fibers exhibit no negative curvature (grey) and one shows negative curvature similar to WT k-fibers (orange). Scatter plot of microneedle positions shown above (inset). (**F**) Left: Distribution of microneedle positions along the k-fiber in WT (n = 17 cells) and PRC1 RNAi (n = 12 cells) spindles, after datasets are minimally down-sampled to maximize microneedle position overlap between them. There is no significant difference in microneedle position in the two conditions (p=0.38, Mann-Whitney U test). Plot shows mean ± SEM. Right: Percentage of deformed k-fiber profiles showing negative curvature near chromosomes in WT and PRC1 RNAi manipulated spindles, showing loss of negative curvature without PRC1. (**G**) Schematic of the three measurements made in (**H,I,J**): Inter-kinetochore distance between the manipulated k-fiber and its sister, angle between the sister k-fiber plus-end (opposite the manipulated k-fiber) and the pole-pole axis, and the angle between sister k-fiber plus-end regions. (**H**) Change in inter-kinetochore distance in WT unmanipulated (control, n = 13 kinetochore pairs from 6 cells), WT manipulated (n = 8 kinetochore pairs from 8 cells) and PRC1 RNAi manipulated (n = 8 kinetochore pairs from 8 cells) spindles, measured over 60 s. Inter-kinetochore distance after manipulation is significantly higher in spindles with PRC1 RNAi than WT (p=0.003, Mann-Whitney U test). Plot shows mean ± SEM. (**I**) Change in angle of sister k-fiber plus-end with respect to the pole-pole axis in WT unmanipulated (control, n = 12 k-fibers from 6 cells) and WT manipulated (n = 11 k-fibers from 11 cells) and PRC1 RNAi manipulated (n = 9 k-fibers from 9 cells) spindles, measured over 60 s. The sister k-fiber moves less (smaller angle) towards the pole-pole axis after manipulation in PRC1 RNAi spindles compared to WT (p=0.004, Mann-Whitney U test). Plot shows mean ± SEM. (**J**) Distribution of the angle between sister k-fiber plus-end regions at the end of manipulation in WT (n = 20 cells) and PRC1 RNAi (n = 10 cells) spindles, measured between the chromosome-proximal regions of the k-fibers.

The online version of this article includes the following video, source data, and figure supplement(s) for figure 4:

**Source data 1.** This spreadsheet contains the fluorescence intensity ratio of PRC1 to tubulin along the pole-pole axis of spindles acquired by immuno-fluorescence (*Figure 4B*), the local curvature along k-fibers manipulated over 60 s in PRC1 RNAi spindles (*Figure 4E*), microneedle positions from 60 s manipulations in WT and PRC1 RNAi such that their positions along the k-fiber maximally overlap (*Figure 4F*), the change in inter-kinetochore distance (*Figure 4H*) and angle of sister k-fiber plus-end from the pole-pole axis (*Figure 4I*) in unmanipulated and manipulated spindles, and PRC1 RNAi manipulated spindles, and the angle between sister k-fiber plus-end regions in WT and PRC1 RNAi PtK2 spindles (*Figure 4J*).

**Figure supplement 1.** Validation of PRC1 depletion by RNAi.

**Figure supplement 1—source data 1.** This spreadsheet contains the fluorescence intensity of PRC1 (normalized to background levels) in PtK2 mock RNAi and PRC1 RNAi spindles from immunofluorescence images (*Figure 4—figure supplement 1C*).

**Figure supplement 2.** Immunofluorescence quantifications of inter-kinetochore distance and tubulin intensity between PRC1 RNAi and mock RNAi spindles.

**Figure supplement 2—source data 1.** This spreadsheet contains the inter-kinetochore distance of mock RNAi and PRC1 RNAi spindles (*Figure 4—figure supplement 2A*), and the fluorescence intensity of tubulin (normalized to background levels) in mock RNAi and PRC1 RNAi spindles (*Figure 4—figure supplement 2B*) in PtK2 cells.

**Figure 4—video 1.** The microtubule crosslinker PRC1 mediates the specialized and short-lived kinetochore-fiber reinforcement near chromosomes.
https://elifesciences.org/articles/53807#fig4video1

To assess whether PRC1 played a role in this specialized reinforcement, we depleted it by RNAi in PtK2 cells (*Figure 4C*, *Figure 4—figure supplement 1*; *Udy et al., 2015*). Using immunofluorescence, we observed a decrease in the inter-kinetochore distance (from 2.35 ± 0.04 μm in WT cells to 1.86 ± 0.09 μm in PRC1 RNAi cells) (*Figure 4—figure supplement 2A*), similar to human cells (*Polak et al., 2017*). When subjecting PRC1 RNAi spindles to the same manipulation assay (deformed by 2.6 ± 0.1 μm over 66 ± 6 s (n = 12 cells)) as WT spindles (*Figure 3*), we found that 91% of the spindles lacked a detectable k-fiber negative curvature near chromosomes upon pulling (*Figure 4D–E*, *Figure 4—video 1*). In order to directly compare the WT and PRC1 RNAi datasets, we only looked at curvature profiles in which the distribution of microneedle positions overlapped: 76% (n = 13/17) of deformed k-fibers in WT cells showed negative curvature near chromosomes compared to only 8% (n = 1/12) of deformed k-fibers in PRC1 RNAi cells (*Figure 4F*). Thus, PRC1 laterally reinforces k-fibers near chromosomes, and enables them to resist pivoting under force.

In PRC1 RNAi spindles, the inter-kinetochore distance increased by 0.7 ± 0.1 μm upon manipulation, compared to 0.1 ± 0.1 μm in WT spindles (*Figure 4G,H*). This suggests that PRC1-mediated crosslinking not only resists compressive forces (*Kajtez et al., 2016*) but also extensile forces. Furthermore, upon pulling, the sister k-fiber opposite the deformed k-fiber moved in towards the pole-pole axis by 4.5 ± 1.4˚, less than in WT cells where it moved by 10.6 ± 1.2˚ (*Figure 4G,I*). Finally, the angle between the sister k-fibers' chromosome-proximal regions was less well-preserved after manipulation than in WT (*Figure 4G,J*). These findings suggest that PRC1 promotes tight coupling

between sister k-fibers, ensuring they behave as a single mechanical unit and maintain biorientation. While we do not know if PRC1 acts directly or indirectly to locally reinforce k-fibers, we find that microtubule intensity remains similar upon PRC1 depletion (*Figure 4—figure supplement 2B*), suggesting that it is unlikely to act simply by changing microtubule density in the spindle. Together, our findings indicate that PRC1 provides the specialized, short-lived structural reinforcement in the spindle center near chromosomes. They suggest a model whereby PRC1 can locally protect the spindle center from transient lateral and longitudinal forces while allowing it to move chromosomes and remodel over longer timescales (*Figure 5*).

## Discussion

The spindle's ability to be dynamic and constantly generate and respond to force while robustly maintaining its structure is central to chromosome segregation. Here, we asked: what mechanisms allow the dynamic mammalian spindle to robustly hold on to its k-fibers? Using microneedle manipulation to directly pull on k-fibers, we were able to challenge the robustness of their anchorage over different locations and timescales (*Figure 1*). We show that k-fibers' anchorage in the spindle is local in space and short-lived in time (*Figure 5*). K-fibers are weakly coupled to their neighbors but strongly coupled to their sisters (*Figure 2*) through specialized, short-lived reinforcement within the first 3 μm of chromosomes (*Figure 3*), mediated by PRC1 (*Figure 4*). Such mechanical reinforcement could help protect chromosome-to-spindle connections while allowing them to remodel over the course of mitosis. Together, our work provides a framework for understanding the molecular and physical mechanisms giving rise to the dynamics and robust mechanics of the mammalian spindle.

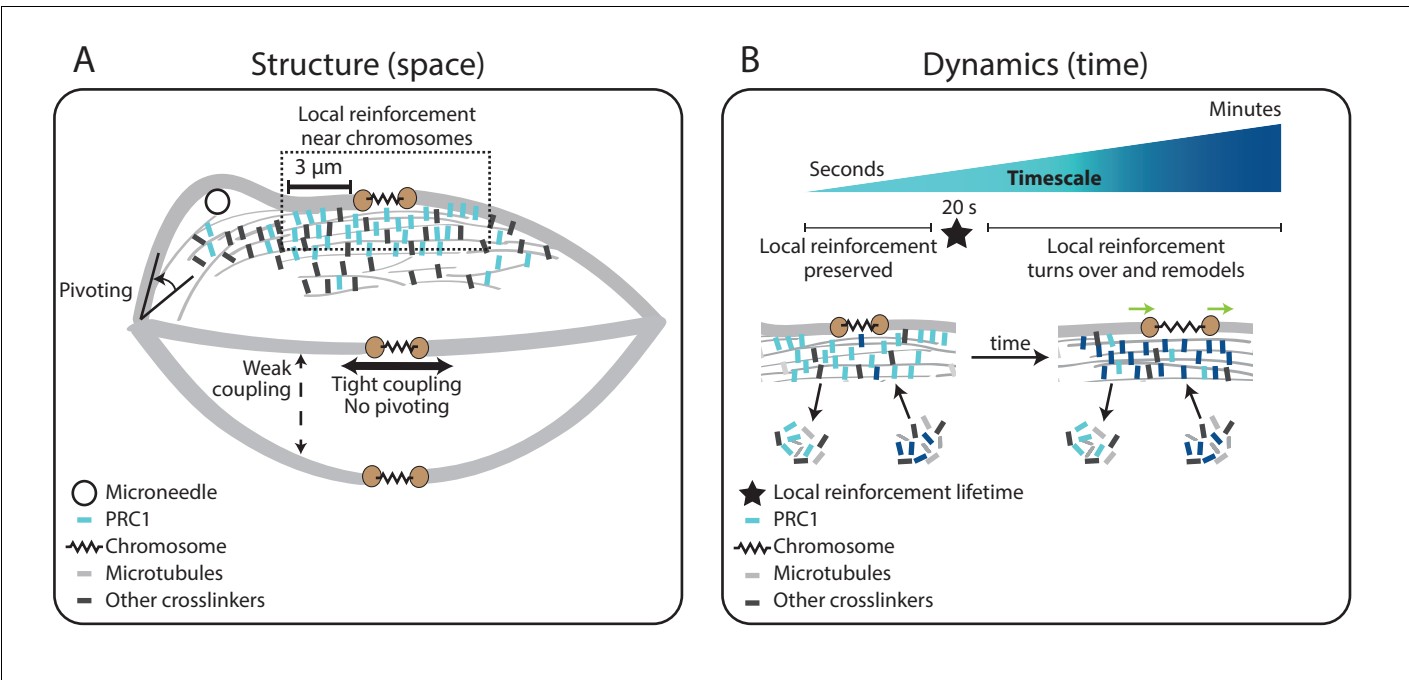

**Figure 5.** Model for specialized, short-lived reinforcement near chromosomes in the mammalian spindle. (**A**) K-fiber reinforcement in space: Microneedle (black circle) manipulation of the mammalian spindle reveals that k-fibers (light grey) are weakly coupled to their neighbors (thin dashed vertical line), strongly coupled to their sisters (thick horizontal line), and pivot around the pole (black arrow) but not around chromosomes. K-fibers are locally reinforced (dashed box, 3 μm) near chromosomes (spring) through specialized, non-uniform mechanisms requiring the microtubule crosslinker PRC1 (light blue squares). Other crosslinkers in the spindle are shown in dark grey. (**B**) K-fiber reinforcement in time: Local reinforcement near chromosomes is preserved over seconds (with a lifetime of 20 s, black star) yet remodels over minutes as molecules turn over in the spindle center. This allows the local architecture in the spindle center to persist under transient force fluctuations, and yet respond to sustained forces. Such short-lived reinforcement could help protect chromosome-to-spindle connections while allowing them to remodel (green arrow) as mitosis progresses. For simplicity, we only depict PRC1 turning over as time evolves (from light blue to dark blue PRC1 molecules), though microtubules and other crosslinkers also turn over.

Spindle mechanics emerge from both active (energy consuming) and passive molecular force generators (*Elting et al., 2018*). Here, we find a spatially and temporally well-defined role for PRC1, a passive molecular force generator that preferentially binds anti-parallel microtubules (*Mollinari et al., 2002*) and can maintain and reinforce microtubule overlaps *in vitro* (*Bieling et al., 2010*; *Braun et al., 2011*; *Wijeratne and Subramanian, 2018*). In the *longitudinal axis*, PRC1 can help promote chromosome stretch, which has been proposed to maintain tension and biorientation of sister-kinetochore pairs (*Polak et al., 2017*). We find that PRC1 can also help limit chromosome stretch (*Figure 4H*), thereby mechanically buffering chromosomes from transient forces. Whether PRC1 mechanically reinforces the spindle center directly or indirectly remains unknown; for example, it could do so by generating friction along microtubules (*Forth et al., 2014*), thereby limiting the timescale of microtubule sliding and spindle reorganization. PRC1's microtubule binding is phosphoregulated during mitosis (*Mollinari et al., 2002*; *Zhu and Jiang, 2005*; *Zhu et al., 2006*), and as such these frictional forces may be regulated as the spindle's mechanical functions change. In the *lateral axis*, we find that PRC1 restricts pivoting of k-fibers around chromosomes (*Figure 4D-E*), even under external force, thereby promoting biorientation between sister k-fibers. Whether PRC1 mediates this mechanical reinforcement by crosslinking k-fibers to a non-specific or specific set of non-kMTs (such as bridge-fibers *Kajtez et al., 2016*; *Polak et al., 2017*), and whether PRC1 plays the same mechanical role over different timescales or spindle axes (*Elting et al., 2017*), remain open questions. Looking forward, defining the mechanical roles of diverse crosslinkers such as NuMA (*Elting et al., 2017*) and Kinesin-5 (*Shimamoto et al., 2015*; *Takagi et al., 2019*), combined with the use of calibrated microneedles (*Nicklas, 1983*; *Shimamoto and Kapoor, 2012*; *Shimamoto et al., 2011*; *Takagi et al., 2019*), will allow us to quantitatively link molecular-scale mechanics to cellular-scale ones in the mammalian spindle.

Mapping mechanics over space, microneedle manipulation reveals that anchorage along k-fibers is non-uniform and locally reinforced near chromosomes in both the longitudinal and lateral axes (*Figure 3*). While the spindle was known to be able to bear load locally in space (*Elting et al., 2017*; *Milas and Tolić, 2016*), whether anchorage was uniform along the k-fiber's length, and along which axes it acted, were not known. Our work suggests a strategy whereby anchoring forces are spatially mapped to regions of active force generation at the kinetochore (*Grishchuk, 2017*; *Inoué and Salmon, 1995*; *Mitchison et al., 1986*) and spindle poles (*Elting et al., 2014*; *Sikirzhytski et al., 2014*), similar to patterns in *Xenopus* extract spindles (*Takagi et al., 2019*) despite significant differences in spindle architecture (*Crowder et al., 2015*). Probing mechanical heterogeneity in other regions in the spindle will further enable us to map local mechanical properties to function. In principle, specialized reinforcement near chromosomes could help protect kinetochore-microtubule attachments and chromosomes from transient forces, and ensure that sister k-fibers point to opposite poles (*Figure 2E–H*) – while allowing them to bend further away and focus into poles.

Mapping mechanics over time, our findings indicate that local anchorage at the spindle center can robustly resist structural changes due to forces over seconds (*Figure 2F*, *Figure 3*), and yet remodel over minutes. Our findings suggest that spindle structural and functional robustness emerge in part from differentially responding to forces over different timescales. While the molecular basis of the spindle center's remodeling timescale is not known, it likely reflects the turnover of underlying connections, for example of non-kMTs (*Saxton et al., 1984*) or of PRC1 (*Pamula et al., 2019*; *Subramanian et al., 2010*), just as crosslinking dynamics contribute to the physical properties of *Xenopus* extract spindles (*Shimamoto et al., 2011*). By tuning the lifetime of these connections that reinforce the spindle center, the cell could in principle regulate its remodeling to allow kinetochores to only sense forces of a given timescale. This could, for example, result in only sustained forces being communicated to kinetochores (*Long et al., 2019*), thereby ensuring that the error correction machinery responds to the appropriate mechanical cues (*Li and Nicklas, 1995*; *Sarangapani and Asbury, 2014*). Furthermore, regulating the timescale of remodeling can enable spindle morphology to change only when needed, for instance at the metaphase-to-anaphase transition (*Zhu and Jiang, 2005*; *Zhu et al., 2006*). Looking forward, the ability to exert controlled forces on the mammalian spindle will be key to understanding how its mechanics emerge (*Brugués and Needleman, 2014*; *Oriola et al., 2018*) from the dynamics of its individual components (*Roostalu et al., 2018*; *Ross et al., 2019*; *Surrey et al., 2001*).

Altogether, our work suggests that mechanical heterogeneity is a simple principle for how the spindle and other macromolecular machines can be at once dynamic and mechanically robust.

Mechanical heterogeneity over space and time allows these machines to be reinforced in specific regions and on short timescales for local functions, while allowing remodeling elsewhere and on longer timescales – to ultimately perform robust cellular-scale functions.

## Materials and methods

### Key resources table

| Reagent type (species) or resource | Designation | Source or reference | Identifiers | Additional information |
|---|---|---|---|---|
| Cell line (P. tridactylus) (male) | PtK2 GFP-tubulin | A. Khodjakov | PMID:12604591 | kidney epithelial, stably expressing GFP-α-tubulin |
| Antibody | Rabbit anti-PRC1 (H-70) | Santa Cruz Biotechnologies | Cat# sc-8356 | IF (1:100), RRID:AB_2169665 |
| Antibody | Mouse-anti-α-tubulin | Sigma-Aldrich | Cat# T6199 | IF (1:1000), WB(1:5000), RRID:AB_477583 |
| Antibody | Mouse anti-PRC1 | Biolegend | Cat# 629001 | WB (1:300), RRID:AB_2169532 |
| Antibody | Alexa 488 goat anti-mouse IgG | Invitrogen | Cat# A11001 | IF (1:500), RRID:AB_2534069 |
| Antibody | Alexa 647 goat anti-rabbit IgG | Life Technologies | Cat# A21244 | IF (1:500), RRID:AB_141663 |
| Antibody | Goat anti-mouse IgG-HRP | Santa Cruz Biotechnologies | Cat# sc-2055 | WB (1:10000), RRID:AB_631738 |
| Chemical compound, drug | Oligofectamine | Life Technologies, Carlsbad, CA | Cat# 12252011 | |
| Chemical compound, drug | FCPT | T. Mitchison | PMID:18559893 | 200 µM |
| Chemical compound, drug | BSA Alexa Fluor 555 conjugate | Invitrogen | Cat# A34786 | PMID:22653161 |
| Chemical compound, drug | BSA Alexa Fluor 647 conjugate | Invitrogen | Cat# A34785 | PMID:22653161 |
| Chemical compound, drug | Propidium Iodide | Thermo Fisher | Cat# P3566 | 25 µg/mL |
| Chemical compound, drug | CellMask-Orange | Thermo Fisher | Cat# C10045 | 5 µg/mL |
| Sequence-based reagents | siRNA against PtK PRC1: | Sigma | PMID:26252667 | 5'-GGACTGAGGUUGUCAAGAA-3' |
| Software, algorithm | Python | | 2.7 and 3.0 | |
| Software, algorithm | FIJI | | PMID:22743772 | |
| Software, algorithm | Metamorph | MDS Analytical Technologies | 7.10.3 | |

### Cell culture and siRNA

All work herein was performed using PtK2 GFP-α-tubulin cells. PtK2 GFP-α-tubulin cells were cultured in MEM (11095; Thermo Fisher, Waltham, MA) supplemented with sodium pyruvate (11360; Thermo Fisher), non-essential amino acids (11140; Thermo Fisher), penicillin/streptomycin, and 10% heat-inactivated fetal bovine serum (10438; Thermo Fisher). The cell line tested negative for

mycoplasma, and while we did not authenticate it, its cell behavior and growth characteristics are similar to those reported for the parental PtK2 cell line, whose transcriptome we sequenced (*Udy et al., 2015*). Cells were maintained at 37°C and 5% $CO_2$. For depletion of PRC1, cells were transfected with siRNA (5'-GGACTGAGGUUGUCAAGAA-3') for PRC1 using Oligofectamine (Life Technologies, Carlsbad, CA) as previously described (*Udy et al., 2015*). Cells were imaged 72 hr after siRNA treatment. For PRC1 RNAi, knockdown was validated by western blot and immunofluorescence analyses, and counting binucleated cells. Western blot analysis showed an 88% decrease of PRC1 levels in PRC1 RNAi vs. mock RNAi (Luciferase) conditions, after normalizing to tubulin levels (*Figure 4—figure supplement 1A*). By immunofluorescence, the average per pixel PRC1 intensity in the spindle above that in the cytoplasm was 597.1 ± 13.7 (AU) (SEM, n = 12) in mock RNAi and 145.4 ± 22.3 (AU) (SEM, n = 24) in PRC1 RNAi cells (76% knock-down, *Figure 4—figure supplement 1C*). Cells used for analysis were selected based on examining the DNA channel only (so as to be unbiased for the amount of PRC1 present when selecting cells, as a control for live experiments). We confirmed PRC1 knockdown in the particular coverslips used for live imaging by verifying at low magnification the enrichment of binucleated cells (from 4% (n = 200 cells) in mock RNAi to 35% of cells (n = 213) in PRC1 RNAi) (*Figure 4—figure supplement 1D*), a previously characterized consequence of PRC1 knockdown (*Mollinari et al., 2002*; *Udy et al., 2015*).

## Drug/dye treatment

To image the cell membrane, we added CellMask-Orange (Thermo Fisher) (1:1000 dilution) to the imaging dish 1 min before imaging (*Figure 1A–D*): we observed rapid incorporation of dye into the cell membrane and imaged cells for 30–45 min, before too much membrane dye became internalized.

To test whether the membrane was ruptured during microneedle manipulation, we added propidium iodide (Thermo Fisher, Waltham, MA) (50 µl of 1 mg/ml stock solution) to the cell media 1 min before imaging (*Figure 1—figure supplement 1*): We observed rapid chromosome labeling in dead cells and no labeling of chromosomes in cells that were successfully manipulated.

To increase microtubule crosslinking by rigor binding Eg5, we treated with 200 µM FCPT (2-(1-(4-fluorophenyl)cyclopropyl)−4-(pyridin-4-yl)thiazole) (gift of T Mitchison, Harvard Medical School, Boston, MA) for 15–30 min (*Groen et al., 2008*; *Figure 2D*).

## Immunofluorescence

To quantify the region of PRC1 enrichment in the metaphase spindle in PtK2 cells (*Figure 4A–B*) and confirm PRC1 depletion following RNAi in PtK2 cells (*Figure 4C*, *Figure 4—figure supplement 1B*), cells were fixed with 95% methanol + 5 mM EGTA at −20°C for 1 min, washed with TBS-T (0.1% Triton-X-100 in TBS), and blocked with 2% BSA in TBS-T for 1 hr. Primary and secondary antibodies were diluted in TBS-T+2% BSA and incubated with cells for 1 hr (primary) and for 25 min at room temperature (secondary). DNA was labeled with Hoescht 33342 (Sigma, St. Louis, MO) before cells were mounted in ProLongGold Antifade (P36934; Thermo Fisher). Cells were imaged using the spinning disk confocal microscope described above. Antibodies: rabbit anti-PRC1 (1:100, BioLegend, San Diego, CA), mouse anti-α-tubulin DM1α (1:1000, Sigma-Aldrich), anti-mouse secondary antibodies (1:500) conjugated to Alexa Fluor 488 (A11001; Invitrogen), anti-rabbit secondary antibodies (1:500) conjugated to Alexa Fluor 647 (A21244; Life Technologies).

We observed PRC1 intensity at spindle poles but this intensity is similar in both mock RNAi (Luciferase) and PRC1 RNAi cells (0.80 ± 0.07 in mock RNAi vs. 0.86 ± 0.08 in PRC1 RNAi (AU), n = 20 cells, p=0.61, Mann-Whitney U test), suggesting non-specific antibody labeling by the human PRC1 antibody at spindle poles in PtK2 rat kangaroo cells.

## Western blot

PtK2 cells were transfected with siPRC1 or siLuciferase as a control. Cells were lysed, instead of imaged, 72 hr post-transfection. Samples were separated on a NuPAGE 4–12% Bis-Tris Protein gel (NP0321PK2, Thermo Fisher, Waltham, MA), transferred to a nitrocellulose membrane and blotted with mouse anti-PRC1 (1:1000, BioLegend, San Diego, CA) and anti-tubulin DM1α (1:5000, Sigma, St. Louis, MO) as a loading control. The membranes were probed with horseradish peroxidase-

conjugated secondary antibodies (1:10000, Santa Cruz Biotechnology, Santa Cruz, CA). Images were quantified using ImageJ.

## Imaging

PtK2 GFP-α-tubulin cells (stable line expressing human α-tubulin in pEGFP-C1, Clontech Laboratories, Inc; a gift from A Khodjakov, Wadsworth Center, Albany, NY; *Khodjakov et al., 2003*) were plated on 35 mm #1.5 coverslip glass-bottom dishes coated with poly-D-lysine (MatTek, Ashland, MA) and imaged in $CO_2$-independent MEM (Thermo Fisher). The cells were maintained at 27–32°C in a stage top incubator (Tokai Hit, Fujinomiya-shi, Japan), without a lid. Live imaging was performed on two similar CSU-X1 spinning-disk confocal (Yokogawa, Tokyo, Japan) Eclipse Ti-E inverted microscopes (Nikon) with a perfect focus system (Nikon, Tokyo, Japan). The 12 s manipulations (*Figure 2*) were performed on a microscope with the following components: head dichroic Semrock Di01-T405/488/561, 488 nm (150 mW) and 561 (100 mW) diode lasers (for tubulin and microneedle respectively), emission filters ETGFP/mCherry dual bandpass 59022M (Chroma Technology, Bellows Falls, VT), and Zyla 4.2 sCMOS camera (Andor Technology, Belfast, United Kingdom). The 60 s manipulations (*Figures 3* and *4*) and immunofluorescence (*Figure 4*) were performed on a microscope with the following components: head dichroic Semrock Di01-T405/488/568/647, 488 nm (120 mW) and 642 nm (100 mW) diode lasers (for tubulin and microneedle respectively), emission filters ET 525/50M and ET690/50M (Chroma Technology), and iXon3 camera (Andor Technology). Cells were imaged via Metamorph (7.10.3, MDS Analytical Technologies) by fluorescence (50–70 ms exposures) with a 100 × 1.45 Ph3 oil objective through a 1.5X lens yielding 105 nm/pixel at bin = 1. For the 3D whole cell membrane imaging (*Figure 1A–B*), z-stacks were taken through the entire cell with a z step-size of 400 nm. For the 12 s manipulations (*Figure 2*), the camera was used in continuous streaming mode, where single z-plane images where taken every 120 ms, which enabled us to build the strain map more accurately. For the 60 s manipulations (*Figures 3* and *4*), cells were imaged by taking either a single slice or 3 z-slices of 400 nm spacing every 5–7 s, helping us track the deformed k-fiber over time despite z-height changes induced by microneedle movement.

## Microneedle manipulation

Microneedle manipulation was adapted to mammalian cells by optimizing the following key parameters:

- **Making microneedles:** Glass capillaries with an inner and outer diameter of 1 mm and 0.58 mm respectively (1B100-4 or 1B100F-4, World Precision Instruments) were used to create microneedles. A micropipette puller (P-87, Sutter Instruments, Novato, CA) was used to create uniform glass microneedles. When pulled the tip of the capillary was closed, as seen in the microneedle-labeled image in *Figure 1B*. For a ramp value of 504 (specific to the type of glass capillary and micropipette puller), we used the following settings: Heat = 509, Pull = 70, velocity = 45, delay = 90, pressure = 200, prescribed to generate microneedles of 0.2 μm outer tip diameter (Sutter Instruments pipette cookbook). In the plane of imaging, microneedle diameter was measured to be 1.1 ± 0.1 μm. This variability comes from the microneedle tip sometimes being in a slightly different z-plane than the plane imaged. Microneedles with longer tapers and smaller tips than above ruptured the cell membrane more frequently.
- **Bending microneedles:** Microneedles were bent ~1.5 mm away from their tip to a 45° angle using a microforge (Narishige International, Amityville, NY), so as to have them approach the coverslip at a 90° angle (the microneedle holder was 45° from the coverslip). The angle of microneedle approach was critical towards improving cell heath during manipulations, likely because it minimizes the surface area of the membrane and cortex deformed by the microneedle.
- **Coating microneedles:** Microneedles used for manipulation were coated with BSA Alexa Fluor 555 conjugate (BSA-Alexa-555; A-34786, Invitrogen) (*Figure 2*) or BSA Alexa Fluor 647 conjugates (BSA-Alexa-647; A-34785, Invitrogen) (*Figures 1*, *3* and *4*) by soaking in the solution for 60 s before imaging (*Sasaki et al., 2012*): BSA-Alexa dye and Sodium Azide (Nacalai Tesque, Kyoto, Japan) were dissolved in 0.1 M phosphate-buffered saline (PBS) at the final concentration of 0.02% and 3 mM, respectively (*Sasaki et al., 2012*). Tip labeling was critical towards improving cell heath during manipulations because it allowed us to better visualize the microneedle tip in fluorescence along with the spindle and prevented us from going too deeply into the cell, thereby causing rupture.

- **Choosing cells**: Mitotic cells for microneedle manipulation were chosen based on the following criteria: spindles in metaphase, flat, bipolar shape with both poles in the same focal plane. These criteria were important for pulling on single k-fibers close to the top of the cell and simultaneously being able to image the whole spindle's response to manipulation.

- **Mounting and controlling the micromanipulator**: The micromanipulator was mounted to the microscope body using a metal bracket and was positioned just above the microscope stage. Manipulations were performed in 3D using an x-y-z stepper-motor micromanipulator (MP-225, Sutter Instruments, Novato, CA). A 3-axis-knob (ROE-200) or joystick (Xenoworks BRJOY, Sutter Instruments) connected to the manipulator via a controller box (MPC-200, Sutter Instruments) allowed fine manual movements and was used to both find and position the microneedle tip in the field of view and manipulate the spindle while imaging.

- **Finding and positioning microneedles**: The microneedle tip was located and centered in the field of view using low magnification (10X or 20 × 0.5 Ph1 air objectives) imaging. Critically, the microneedle tip was brought down close to the coverslip, placed just above the cells, after which higher magnification (100 × 1.45 Ph3 oil objective) imaging was used to refine the x-y-z position of the microneedle right above the cell. When refining the microneedle position in higher magnification, using the Ph1 phase ring helped see the microneedle more clearly than with a Ph3 ring.

- **Contacting the cell with microneedles:** When starting a manipulation experiment, the microneedle was placed ~5 µm above the cell and images were acquired every 5–7 s. Once an outer k-fiber was identified in a plane near the top of the cell, the microneedle was slowly brought down into the cell, using the fluorescent label of the microneedle tip to help inform on its position. If the microneedle's position were far away from the k-fiber of interest, the microneedle was slowly moved out of the cell, its x-y position adjusted, and it was brought back down into the cell. Through this iterative process, microneedle such that it was inside the spindle, right next to the outer k-fiber.

- **Moving microneedles:** Once the microneedle was positioned next to an outer k-fiber near the top of the cell, it was moved in a direction that was roughly perpendicular to the pole-pole axis. All 12 s manipulations were done manually using the joystick, and most 60 s manipulations were done with computer control (*Source code 1.*) that took the following inputs: Angle of movement (based on the orientation of the spindle in the cell), duration of movement and total distance. The script generates a text file with a sequence of steps (with the smallest step-size being 0.0625 µm) in xyz and wait/delay times, which are the instructions for the software (Multi-Link, Sutter Instruments) that makes the manipulator move. Computer control ensured smooth and reproducible microneedle movements over a longer period. Our manipulation programs generated microneedle movements of the following speeds: 9.3 ± 1.8 µm/min and 2.5 ± 0.1 µm/min. Microneedle speeds that exceeded these killed cells more frequently. For the manipulate-and-hold experiments (*Figure 3H–I*), the microneedle was left in the same position at the end of its movement and only removed after 45–60 s. At the end of the manipulation, the microneedle was manually removed from the cell slowly (<~5 µm/min) to avoid membrane rupture or cell detachment from the coverslip.

- **Objectively selecting cells for analysis:** Cells were included in our datasets if they did not appear negatively affected by micromanipulation. Cells were excluded from the dataset if they met one or more of the following criteria: cells that underwent sudden and continuous blebbing upon microneedle contact, spindles that started to rapidly collapse during manipulation, chromosomes decondensed, mitochondria become punctate. When we followed the spindle post manipulation (n = 10), 70% of cells entered anaphase within 15 min after manipulation.

- **Disrupting spindle structure with microneedles**: Inherent to microneedle manipulation experiments, where the microneedle contacts and dips into the cell, tubulin is locally displaced from the volume that the microneedle tip occupies (*Figure 1A–B*). It is likely that the microneedle locally disrupts microtubules and their crosslinks where it is inserted. However, our interpretation of the data is independent of such local disruption. The facts that we always observed negative curvature between 1 and 3 µm from the chromosome regardless of where we pulled (*Figure 3G*), and that the negative curvature disappeared in the absence of PRC1 (*Figure 4D–F*), indicate that the negative curvature is not simply a result of the needle disrupting the spindle structure, but is the result of a specialized structure in the spindle center reinforcing k-fibers there.

## Quantification and statistical analyses

Building strain maps (*Figure 2C*, *Figure 2—figure supplement 2B*): First, we aligned all images during the manipulation in order to correct for whole spindle rotation and translation, using the Stackreg plugin on ImageJ (*Thévenaz et al., 1998*). This correction allowed us to only look at structural changes within the spindle and not whole spindle translation or rotation. K-fibers were included in the data set (*Figure 2*) only if their entire length stayed within the same z-plane over time. The images taken in the continuous streaming mode (50 frames) helped ensure that the same k-fiber could be followed over a long time period, being correctly mapped from frame to frame. K-fibers were traced and tracked semi-automatically using GFP-α-tubulin images at frames 0 and 50, over their entire length. We stored 100 equally spaced coordinates along each k-fiber in frames 0 and 50, which were then connected to each other (coordinate 1 in frame 0 connects to coordinate 1 in the frame 50, and so on). This approach provided a linear mapping between the undeformed (purple) and deformed (green) spindle image to build a strain map (*Figure 2C*, *Figure 2—figure supplement 2*, *Source code 2*). This linear mapping was possible because k-fiber lengths remained constant during these 12 s manipulations (*Figure 2—figure supplement 1*).

Tracking features of interest in live images (*Figures 2*, *3* and *4*): Inter-kinetochore distance (*Figures 2F* and *4H*, *Figure 3—figure supplement 2B*) was calculated as the distance between sister k-fiber plus-ends. To make sure we measured this distance between correctly identified sister pairs, we confirmed that there were correlated movements between them before and during the manipulation. Angle of sister k-fiber plus-end from the pole-pole axis was calculated by measuring the angle between the position of the sister k-fiber's plus-end (connected to the same chromosomes as the deformed one) to the pole-pole axis (*Figures 2H* and *4I*, *Figure 3—figure supplement 2C*). For the control dataset (unmanipulated spindles), the same measurements were made only on outer k-fibers, in order to be able to compare k-fibers in a similar part of the spindle. Pole-pole distance was calculated as the distance between centroids of the two spindle poles (*Figure 2G*). The number of measurements (n) represent a subset of the manipulation dataset per figure, depending on which features were trackable in the manipulation.

Measuring curvature along k-fiber (*Figures 3* and *4*, *Figure 3—figure supplement 1*): We used a custom-written Python script to calculate local curvature along k-fiber length (*Source code 3*). We calculated the radius of a circle that was fit to three points along the k-fiber. These three points were chosen to be spaced apart by 1 µm in order to calculate curvature on the relevant length scale. This radius (radius of curvature, units = µm) was used to calculate curvature (units = 1/µm) by taking its inverse, which we then mapped on to the traced k-fiber using a color spectrum from blue (negative curvature) to red (positive curvature).

Immunofluorescence quantification: In order to quantify the length of PRC1 enrichment in mock RNAi (control) spindles (*Figure 4A,B*), we calculated the ratio of PRC1 to tubulin intensity inside a region of the spindle (whole spindle excluding spindle poles) (*Figure 4A*, bottom panel). In order to quantify the percentage of PRC1 knocked down, we calculated the per pixel intensity of PRC1 in PRC1 RNAi spindles relative to the background levels and compared it to that in mock RNAi spindles (*Figure 4—figure supplement 1C*). A similar analysis was done to quantify microtubule intensity in mock RNAi and PRC1 RNAi spindles inside two regions (whole spindle excluding spindle poles and just the spindle center) (*Figure 4—figure supplement 2B*). Inter-kinetochore distance in mock RNAi and PRC1 RNAi spindles (*Figure 4—figure supplement 2A*) was calculated as the distance between sister k-fiber plus-ends. We equally sampled outer and middle sister k-fiber pairs.

Statistical tests: We used the non-parametric two-sided Mann-Whitney U test when comparing two independent datasets and the Wilcoxon signed rank test when comparing two paired datasets. In the text, whenever we state a significant change or difference, the p-value for those comparisons were less than 0.05. To evaluate the correlation between two datasets, we used Pearson as well as Spearman correlation functions, in order to test for linear and monotonic relationships respectively. In the figures, we display the exact p-value from every statistical comparison made, and in the legends we state what test was conducted. Quoted n's are described in more detail where mentioned in the text or figure legend, but in general refer to the number of independent individual measurements (biological replicates, e.g., individual k-fibers, sister k-fiber pairs, spindles, manipulations, etc.).

## Script packages

All scripts were written in Python. Pandas was used for all data organization and compilations, Scipy for statistical analyses, Matplotlib and Seaborn for plotting and data visualization as well as Numpy for general use. FIJI was used for movie formatting, immunofluorescence quantification and tracking manipulations (*Schindelin et al., 2012*). In FIJI, StackReg for spindle rigid body motion correction was necessary for building strain maps and MtrackJ was used to track the microneedle over time (*Meijering et al., 2012*; *Thévenaz et al., 1998*).

## Video preparation

Videos show a single spinning disk confocal z-slice imaged over time (*Figure 2—video 1*) or a maximum intensity projection (*Figure 1—video 1*, *Figure 3—video 1*, *Figure 3—video 2*, *Figure 4—video 1*) and were formatted for publication using ImageJ and set to play at 5 frames per second (*Figure 1—video 1*, *Figure 2—video 1*, *Figure 3—video 1*, *Figure 3—video 2*, *Figure 4—video 1*).

## Acknowledgements

We thank Alexey Khodjakov for PtK2 GFP-α-tubulin cells and Timothy Mitchison for FCPT. We are grateful to Le Paliulis for technical advice and key disussions, to Maya Anjur-Deitrich, Charles Asbury, Justin Biel, the Fred Chang Lab, Daniel Fletcher, Jesse Gatlin, Wallace Marshall, Dick McIntosh, Dan Needleman, Adair Oesterle, Yuta Shimamoto, Radhika Subramanian, Iva Tolic and Orion Weiner for helpful discussions, Miquel Rosas Salvans for technical help, Deepak Krishnamurthy and Jason Town for image analysis discussions, the Verkman Lab for the microforge, and Arthur Molines and members of the Dumont Lab for discussions and critical reading of the manuscript. This work was supported by NIH DP2GM119177, NIH 1R01GM134132, NSF CAREER 1554139, NSF 1548297 Center for Cellular Construction, the Rita Allen Foundation and Searle Scholars' Program (SD), NSF Graduate Research Fellowships (PS and AFL) and a UCSF Kozloff Fellowship (AFL).

## Additional information

### Funding

| Funder | Grant reference number | Author |
| --- | --- | --- |
| National Institute of General Medical Sciences | DP2GM119177 | Sophie Dumont |
| National Institute of General Medical Sciences | 1R01GM134132 | Sophie Dumont |
| National Science Foundation | 1554139 CAREER | Sophie Dumont |
| National Science Foundation | 1548297 Center for Cellular Construction | Sophie Dumont |
| Rita Allen Foundation | | Sophie Dumont |
| Chicago Community Trust | Searle Scholars' Program | Sophie Dumont |
| National Science Foundation | Graduate Research Fellowship | Pooja Suresh Alexandra F Long |
| University of California, San Francisco | UCSF Kozloff Fellowship | Alexandra F Long |

The funders had no role in study design, data collection and interpretation, or the decision to submit the work for publication.

### Author contributions

Pooja Suresh, Conceptualization, Data curation, Software, Formal analysis, Funding acquisition, Validation, Investigation, Visualization, Methodology, Writing - original draft, Writing - review and editing; Alexandra F Long, Funding acquisition, Visualization, Writing - review and editing, Method Optimization; Sophie Dumont, Conceptualization, Resources, Supervision, Funding acquisition, Writing - review and editing

## Author ORCIDs

Pooja Suresh ![ID] https://orcid.org/0000-0002-7793-4827
Alexandra F Long ![ID] https://orcid.org/0000-0002-2471-6797
Sophie Dumont ![ID] https://orcid.org/0000-0002-8283-1523

## Decision letter and Author response

Decision letter https://doi.org/10.7554/eLife.53807.sa1
Author response https://doi.org/10.7554/eLife.53807.sa2

## Additional files

### Supplementary files

• Source code 1. This script generates a sequence of steps in the x and y directions used to program the movement of the micromanipulator.

• Source code 2. This script calculates curvature along a tracked k-fiber, used to generate *Figure 3B*, *Figure 3H*, *Figure 4D* and *Figure 3—figure supplement 1A*.

• Source code 3. This script builds strain maps, used to generate *Figure 2C* and *Figure 2—figure supplement 2B*.

• Transparent reporting form

### Data availability

Source data for all main and supplementary figures have been provided.

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
