## [Decision Letter]

**Acceptance summary:**

In this study, the authors exert local forces on spindles in intact mammalian cells using microneedle manipulation. They observe the preservation of local spindle architecture in the spindle center over several seconds as a response to local pulling. They find that sister kinetochore fibers are mechanically coupled. This depends on the presence of PRC, suggesting that antiparallel microtubule connections are important for local reinforcement. This elegant and technically sophisticated work provides insight into the internal mechanics and dynamics of the spindle in mammalian cells.

**Decision letter after peer review:**

Thank you for submitting your article "Microneedle manipulation of the mammalian spindle reveals specialized, short-lived reinforcement near chromosomes" for consideration by *eLife*. Your article has been reviewed by three peer reviewers, and the evaluation has been overseen by a Reviewing Editor and Anna Akhmanova as the Senior Editor. The following individuals involved in review of your submission have agreed to reveal their identity: Stefanie Redemann (Reviewer #1); Yuta Shimamoto (Reviewer #3).

The reviewers have discussed the reviews with one another and the Reviewing Editor has drafted this decision to help you prepare a revised submission.

Summary:

The authors employ micromanipulation to exert highly localized and temporally controlled force within the spindle. This goes beyond a previous study with a more "global" manipulation of spindle orientation by pushing the cell using microneedles (MBoC 2017). Here, the spindle's response is characterized via measurements of K-fiber geometry, distance between sister kinetochores, and centromere positioning/orientation. A strength of this work is that manipulations are sufficiently mild to be reversible as evident from the restoration of spindle morphology upon removal of the microneedle and subsequent initiation of anaphase in most cells. The main finding is that the architecture of the centromere/K-fiber complex resists pulling forces exerted on the fiber several micrometers away. This 'local reinforcement' is short lived and it requires PRC1, a major microtubule cross-linker. The results are consistent with similar observations in *Xenopus* spindles assembled *in vitro*. However, this manuscript is the first to provide actual data on this subject in normal mammalian spindles in situ. All reviewers found this study interesting and well conducted.

Essential revisions:

1) The main criticism of the reviewers concerned a central interpretation of the manuscript. Because the microneedle appears to break lateral interactions between microtubules, the question was raised whether the authors can indeed conclude that under normal conditions local PRC1-mediated reinforcements exist preferentially in the central spindle (and not at the poles). This issue deserves further discussion.

2) Figure 4: PRC1 localization and depletion: A technical concern was the lack of quantification of the PCR1 depletion efficiency. A quantitative estimate should be provided. Immunofluorescence: does PRC1 localize to regions without microtubules? Is it fair to say that PRC1 preferentially localizes to the spindle center or does rather not localize to (parallel) kinetochore microtubules? Why is the PRC1 signal so strong at spindle poles? (Different from e.g., Polak et al., 2017, Pamula et al., 2019) Also, the spindle in Figure 4C looks like it has much less bundled k-fibers compared to control although Figure 4—figure supplement 2B shows similar intensity. It would be helpful if the authors clarify these points.

---

## [Author Response]

Essential revisions:1) The main criticism of the reviewers concerned a central interpretation of the manuscript. Because the microneedle appears to break lateral interactions between microtubules, the question was raised whether the authors can indeed conclude that under normal conditions local PRC1-mediated reinforcements exist preferentially in the central spindle (and not at the poles). This issue deserves further discussion.

Thank you for raising this important point. We agree that it is central to the interpretation of the data, and that it needs explicit discussion in the paper. Indeed, it is likely that the microneedle locally disrupts microtubules and their crosslinks where it is inserted. However, our interpretation of the data is independent of such local disruption. First, if the local disruption around the microneedle were the origin of the k-fiber’s negative curvature upon pulling, the position of the negative curvature should depend on where the needle is and where it locally disrupts structure. This is not what we find: k-fiber negative curvature occurs near chromosomes no matter where we place the microneedle (Figure 3G). To clarify this point, we have now explicitly mentioned it when we present the data and interpretation that there is preferential reinforcement in the spindle center: “This suggests that the negative curvature we observe is not simply the result of local structural disruption caused by the microneedle’s presence. Instead, it strongly supports a model whereby a specialized structure in the spindle center laterally reinforces k-fibers near chromosomes (Model 2).” Second, while we observe this negative curvature in wildtype cells, it disappears when we deplete PRC1 (Figure 4D-E). Thus, the localized negative curvature we observe in the spindle center is not simply due to the needle disrupting the network away from chromosomes, but due to specialized PRC1-mediated activity in the spindle center. We now explicitly mention in the Materials and methods that the needle likely disrupts the microtubule network locally, and the two reasons above about why our conclusion that local PRC1-mediated reinforcement exist preferentially in the spindle center is not simply due to such local disruption.

2) Figure 4: PRC1 localization and depletion: A technical concern was the lack of quantification of the PCR1 depletion efficiency. A quantitative estimate should be provided.

Thank you for raising this other important point. We agree that quantifying PRC1 depletion efficiency is critical to the interpretation of the data. To address this point, we have now included a new supplemental figure (Figure 4—figure supplement 1). Western blot analysis shows that PRC1 protein levels are reduced by 88% under PRC1 RNAi, normalized to tubulin levels (Figure 4—figure supplement 1A). By immunofluorescence, we find a 76% average decrease in PRC1 levels in PRC1 RNAi cells at metaphase (Figure 4—figure supplement 1B, C) and an increase from 4% to 35% in binucleated cells in the PRC1 RNAi background (Figure 4—figure supplement 1D), a hallmark of failed cytokinesis due to the absence of PRC1. Together, these data demonstrate that PRC1 is efficiently depleted by RNAi in our experimental conditions.

Immunofluorescence: does PRC1 localize to regions without microtubules?

In our immunofluorescence images, we observe PRC1 overlapping with tubulin (Figure 4A), with it preferentially localizing in the center of the spindle (Figure 4B). This is consistent previous images of PRC1 in the spindle (Polak et al., 2017; Zhu and Jiang, 2005). The initial images in Figure 4A displayed maximum intensity projection of z planes taken across the spindle, and we have now replaced them to display a sum intensity projection, which more clearly shows PRC1 colocalize with microtubules. Notably, PRC1 appears to be enriched at spindle poles compared to microtubules (Figure 4A, Figure 4—figure supplement 1B). This appears to be non-specific antibody staining as this localization remains in PRC1 RNAi cells while PRC1 localization in the spindle body disappears (Figure 4—figure supplement 1). We now make note of this non-specific PRC1 staining at spindle poles in the Materials and methods, and see the figure displayed in answer to second question below).

Is it fair to say that PRC1 preferentially localizes to the spindle center or does rather not localize to (parallel) kinetochore microtubules?

PRC1 does localize more strongly to the spindle center than to the rest of the spindle (Figure 4B, and prior work). However, the reviewer is correct that its preference may not be for the center per se, but instead for antiparallel microtubules which are enriched in the spindle center. We have clarified the main text to make this clearer:

“We hypothesized that the microtubule crosslinker PRC1 plays this role based on its localization in the spindle center during metaphase (Mollinari et al., 2002), where antiparallel microtubules are present…”

Why is the PRC1 signal so strong at spindle poles? (Different from e.g., Polak et al., 2017, Pamula et al., 2019)

We thank the reviewers for this question. Immunofluorescence images of human spindles in other papers lack PRC1 labeling at poles (Mollinari et al., 2002; Pamula et al., 2019; Polak et al., 2017; Zhu et al., 2006). In our work in rat kangaroo PtK2 cells using the same antibody against human PRC1, we indeed observe PRC1 labelling at spindle poles (Figure 4A). However, this pole labeling appears non-specific as it remains in PRC1 RNAi conditions where PRC1 disappears from the spindle body: quantifying PRC1 intensity at poles in mock RNAi (Luciferase RNAi) vs. PRC1 RNAi spindles show indistinguishable PRC1 intensity at poles (see Author response image 1). One possibility is that the antibody against human PRC1 recognizes a non-PRC1 protein at poles in rat kangaroo cells. To make this point clear, we have added a note in the Materials and methods that reads: “We observed PRC1 intensity at spindle poles but this intensity is similar in both mock RNAi (Luciferase) and PRC1 RNAi cells (0.80 ± 0.07 in WT vs. 0.86 ± 0.08 in PRC1 RNAi (AU), n = 20 cells, p = 0.61, Mann-Whitney U test), suggesting non-specific antibody labeling at the poles.”

Also, the spindle in Figure 4C looks like it has much less bundled k-fibers compared to control although Figure 4—figure supplement 2B shows similar intensity. It would be helpful if the authors clarify these points.

We went back to the images in Figure 4C, and quantified tubulin intensity in the spindle. We found that those images were not representative of the average tubulin intensity in our dataset. We have replaced Figure 4C with a spindle whose tubulin intensity is much closer to the reported average value in Figure 4—figure supplement 2B.